# Mesenchymal Stem Cell-Derived Extracellular Vesicle-Based Therapy for Alzheimer’s Disease: Progress and Opportunity

**DOI:** 10.3390/membranes11100796

**Published:** 2021-10-19

**Authors:** Yi-An Chen, Cheng-Hsiu Lu, Chien-Chih Ke, Ren-Shyan Liu

**Affiliations:** 1Department of Nuclear Medicine, Cheng Hsin General Hospital, Taipei 112, Taiwan; yachen0414@gmail.com; 2Molecular and Genetic Imaging Core/Taiwan Mouse Clinic, National Comprehensive Mouse Phenotyping and Drug Testing Center, Taipei 112, Taiwan; 3Core Laboratory for Phenomics and Diagnostics, Kaohsiung Chang Gung Memorial Hospital, Kaohsiung 833, Taiwan; rocket2350@yahoo.com.tw; 4Department of Medical Research, Kaohsiung Chang Gung Memorial Hospital, Kaohsiung 833, Taiwan; 5Department of Medical Imaging and Radiological Sciences, Kaohsiung Medical University, Kaohsiung 807, Taiwan; ccke@kmu.edu.tw; 6Department of Medical Research, Kaohsiung Medical University Hospital, Kaohsiung 807, Taiwan; 7Drug Development and Value Creation Research Center, Kaohsiung Medical University, Kaohsiung 807, Taiwan; 8Department of Biomedical Imaging and Radiological Sciences, National Yang Ming Chiao Tung University, Taipei 112, Taiwan; 9National PET and Cyclotron Center (NPCC), Department of Nuclear Medicine, Taipei Veterans General Hospital, Taipei 112, Taiwan

**Keywords:** Alzheimer’s disease, mesenchymal stem cells, extracellular vesicles, therapy

## Abstract

Alzheimer’s disease (AD), as a neurodegenerative disorder, is characterized by mass neuronal and synaptic loss and, currently, there are no successful curative therapies. Extracellular vesicles (EVs) are an emerging approach to intercellular communication via transferring cellular materials such as proteins, lipids, mRNAs, and miRNAs from parental cells to recipient cells, leading to the reprogramming of the molecular machinery. Numerous studies have suggested the therapeutic potential of EVs derived from mesenchymal stem cells (MSCs) in the treatment of AD, based on the neuroprotective, regenerative and immunomodulatory effects as effective as MSCs. In this review, we focus on the biology and function of EVs, the potential of MSC-derived EVs for AD therapy in preclinical and clinical studies, as well as the potent mechanisms of MSC-derived EVs actions. Finally, we highlight the modification strategies and diagnosis utilities in order to make advance in this field.

## 1. Introduction

Alzheimer’s disease (AD) is the world’s most common cause of dementia that will affect over 100 million people by 2050, and which will bring a significant physical, psychological, social and economic burden to patients, their families, caregivers and society [1]. As a neurodegenerative disease, the clinical symptoms of AD include severe cognitive impairments, irreversible memory loss and motor abnormalities, which are attributed to the loss of synapses and neurons in vulnerable regions [2]. AD is characterized by increased neuritic (senile) plaques composed of β-amyloid (Aβ) peptides [3]. Excess aggregated Aβ peptide is generally considered to initiate the pathogenic cascade, including propagation of microtubule-associated tau aggregation throughout the brain [4]. In the past decades, strategies targeting Aβs are mainstream approaches for the treatment and prevention of AD; most of the relevant clinical trials have been conducted at the early/pre-symptomatic stage of AD [5,6]. For instance, the initial trial of aducanumab, an Aβ-directed monoclonal antibody, has shown that it could significantly slow cognitive decline in patients with early stages of AD and reduce Aβ plaques in a dose-and time-dependent manner [7]. Additionally, aducanumab has been approved for medical use in the United States by the FDA in June 2021, but this decision is still controversial and follow-up study is required [8,9]. When it comes to Aβ-targeting drugs, most of them did not show positive outcomes in their phase III trials, e.g., semagacestat, verubecestat, solanezumab and gantenerumab [10,11,12]. Despite that there are five FDA-approved medications for clinical use in dementia, including three cholinesterase inhibitors (donepezil, rivastigmine, and galantamine), a N-methyl-d-aspartate (NMDA) receptor inhibitor (memantine), and a combination therapy with the cholinergic and glutamatergic inhibitors, the symptoms of AD may be improved but the disease progression fails to be halted [1]. It is apparent that a single remedy targeting Aβs is not sufficient to cure AD and the optimal therapeutic approach should tackle Aβ-induced AD pathology as well as prevent cognitive decline simultaneously.

In recent years, mesenchymal stem cells (MSCs) have been used as potential therapeutic cells in multiple diseases due to their immunomodulatory and tissue regenerative properties [13]. MSCs are adult multipotent stem cells that exist in multiple tissues, including bone marrow, adipose tissue, umbilical cord and peripheral blood. They are able to self-renew and differentiate into osteogenic, chondrogenic, adipogenic, myogenic, or stromal lineages. Under different culture conditions, MSCs are reported to differentiate into neuronal cells, hepatocytes, cardiomyocytes, alveolar and gut epithelial cells, making them a promising source in the regenerative medicine. Numerous reports have addressed the beneficial effects of MSCs in damaged tissue repair, including liver failure rescue [14,15], cardiovascular regeneration [16,17], treatment of stroke [18], spinal cord injury [19] and lung fibrosis [20]. Since the characteristics of AD include mass loss of synapses and neurons, MSC transplantation is a rational therapeutic strategy for regeneration of neuronal circuits [21]. Studies have indicated that MSCs are able to reduce Aβ deposition, enhance neurogenesis, alleviate spatial learning and memory deficits in both cellular models and animal models of AD [22,23,24]. Notably, these therapeutic effects in tissue protection and repair are attributed to the paracrine action of MSCs, and further emphasize the role of soluble factors including extracellular vesicles (EVs) secreted from MSCs [25,26].

Several published reviews have described the biogenesis and methodology of isolation of EVs in detail [27,28,29,30,31,32]. In the following paragraphs, we review the origins and characterization of isolated EVs, summarize the current applications of MSC-derived EVs in AD treatments and the molecular/cellular mechanisms of MSC derived EVs actions during therapy, and discuss the potential of drug delivery vehicles and diagnosis utilities for AD. The electronic searches were performed in PubMed, EMBASE, Google Scholar, Clinical Trials database, from 2002 to 2021. The following combinations were used in a search of titles and abstracts in September 2021: Alzheimer’s disease and mesenchymal stem cells; Alzheimer’s disease and mesenchymal stem cells and extracellular vesicles; Alzheimer’s disease and mesenchymal stem cells and exosomes; Alzheimer’s disease and mesenchymal stem cells and microvesicles. The abstracts of all the relevant articles were reviewed by the authors, who further ensured these relevant articles were included in the current review.

## 2. Origins, Classification and Nomenclature of EVs

In general, EVs can be divided into three classes depending on their size and origins, including exosomes, microvesicles (MVs) and apoptotic bodies (ABs) [33]. Exosomes are nanoscale vesicles (30~200 nm) secreted from most types of cells, and commonly found in plasma, tears, urine, breast milk and body fluids [34]. When molecules are transported through the cell membrane via endocytosis, the cargos are formed and then delivered to early endosomes. During the maturation of early endosomes, the cargos are sorted to form interluminal vesicles (ILVs) through the folding back of the endosomal-limiting membrane. ILVs are the origin of exosomes encapsulated by multivesicular bodies (MVBs); the release of ILVs in the form of exosomes is switched in the absence of recycling molecules, such as transferrin receptors or mannose 6-phosphate receptors [35,36].

Different from exosomes, MVs are heterogeneous vesicles with a broad range of size distribution (45–1000 nm), directly budding from the plasma membrane, then released into the extracellular space [37]. The biogenesis of MVs is dependent on the phospholipid asymmetry, lipid transporter activity and calcium signaling [38]. In contrast, the biogenesis of exosomes occurs via the endosomal sorting complex required for a transport (ESCRT)-dependent pathway that is required for the process of cargo ubiquitination in the pre-MVB/early endosomes. The cargos transferred to endosome are selected by the ESCRT-binding ubiquitinated proteins [39]. Despite that the biogenesis of MVs has been reported to be regulated by ESCRT independent pathways, the ESCRT-I subunit TSG101 is found to interact with arrestin domain-containing protein 1 (ARRDC1) to control the release of MVs. In other words, ESCRT proteins are involved in MVs biogenesis [40]. Unlike the canonical ESCRT pathway, a recent report has addressed that the biogenesis of exosomes can be modulated by active RAB31, a Ras-related protein, driving the formation of epidermal growth factor receptor (EGFR)-containing ILV, thereby decreasing GTPase Rab7 activity to prevent the fusion of MVBs with lysosomes and eventually promoting exosomes release [41]. There are various factors involved in biogenesis pathways and the process remains elusive. In addition, apoptotic bodies (ABs), one of EV subtypes, are released when cells undergo apoptosis. Once the apoptosis is induced, a series of events happen, including cell shrinking, chromatin condensation, organelles collapse, and membrane blebbing, leading to the formation of apoptotic bodies (ABs) [42,43]. ABs also carry various molecules, e.g., proteins, lipids, and RNAs, but the size (1–2 μm) is much larger than other EV subgroups [44]. Smaller ABs, termed apoptotic vesicles, have been identified [45]. In fact, ABs present phosphatidylserine (PS) on their surfaces, so they are cleared quickly. It elucidates that the more precise classification of EVs subtypes is needed.

Over the past decade, the number of publications about EVs research is increasing exponentially [46]. These studies include the basic research (biogenesis, secretion, uptake, and pharmacokinetic properties), biomarker identification (EVs, EV-carried proteins, EV-carried RNA, EV-carried DNA, EV-carried microRNA, and EV-carried lipids), pharmaceutical agents (native and engineered EVs), and biomaterial-based drug delivery (loading with protein, microRNA, or drugs) [47,48]. However, the nomenclature of EVs was not defined accurately. These irregular terms used to describe vesicles not only have led to misunderstanding for readers but also caused the findings hard to verify. In 2014, the International Society for Extracellular Vesicles (ISEV) promulgated guidelines called MISEV (minimal information for studies of extracellular vesicles) for the investigators who conduct the EVs studies [47], and now also updated to the latest version (MISEV 2018) [49]. According to these guidelines, “extracellular vesicle” is an expert consensus term and is used to describe the vesicle that cannot replicate and naturally be secreted from the cell and consists of lipid bilayers. Nevertheless, other terms are not prohibited. For instance, the term “exosomes” is generally used in industry, which might fascinate costumers to purchase the related products [50]. In general, the term “EV” is widely used in research and the use of other terms should be defined carefully and clearly.

## 3. Recommendations in Characterization of EVs

Based on MISEV 2018, the characterization of protein markers should be verified using at least three positive markers (one transmembrane/lipid-bound protein is included) and one negative marker. In terms of characterization of a single particle, one electron or atomic force microscopy and one single particle analyzer should be included to examine the size, distribution, and morphology of EVs. Regarding the characterization of protein markers, tetraspanins (e.g., CD9, CD63, CD81, CD82), MVB biogenesis-related protein (Alix, and TSG101), and heat-shock proteins (Hsp60, Hsp70, and Hsp90) are generally used as EVs’ markers [51]. In addition, the proteins specially expressed in cells need to be found in EVs (as cell-type fingerprint) [52]. For example, EVs secreted from T cells and B cells contain T cell receptor (TCR) and B cell receptor (BCR), respectively [53]. MSC-derived EVs bear the markers of CD29, CD44, CD73 and CD90; these markers are widely used in characterization of MSCs [54].

As a subtype of EVs, MVs also package the abundant proteins and nucleic acids that serve as biomarkers for identifying the disease types and the prognosis of disease state [55]. However, the specific markers of MV are still lacking; it is hard to differentiate the MVs from other subgroups of EVs by proteins markers. Since the results obtained from current studies have shown that MVs can be used as good biomarkers for diseases without excluding the existence of other vesicles, pure MVs seem not to be necessary in clinical analysis [56,57,58]. With the large size, ABs can be easily distinguished from smaller vesicles by transmission electron microscopy (TEM), dynamic light scattering (DLS) and nanoparticle tracking analysis (NTA). The specific proteins related to apoptotic process, such as tubulin β-1 and β-4, integrin β-3, Ras-related protein, Fructose-2-P-Aldolase and Glutathione-S-Transferase omega-1, are found in ABs [59]. However, increasing studies have indicated that ABs are not the only secreted vesicles during apoptosis, both apoptotic exosomes (ApoExos) and apoptotic microvesicles (ApoMVs) are also released from apoptotic cells but exhibit different origins, heterogeneities and physical characteristics. Thus, the terms “ApoEVs” are gradually used to describe the EVs released from dying cells rather than “ABs” [60,61]. Collectively, the work of characterizing EVs is under development, and investigators can follow the guidelines recommended by MISEV 2018 to characterize the vesicles they collect.

## 4. Application of MSC-Derived EVs in AD Treatment

EVs act as mediators of intercellular communication through transferring bioactive molecules such as proteins, lipids, mRNAs, microRNAs (miRNAs), genomic DNA and mitochondrial DNA. When recipient cells uptake these bioactive molecules, the molecular machinery of cells is altered in an epigenetic way [51,62,63]. Besides, EVs can serve as drug delivery vectors to transfer enhanced therapeutic agents through chemically or biologically engineering to treat diseases or halt disease progression. The beneficial effect of MSC-derived EVs has been demonstrated in animal models of multiple diseases, such as chronic kidney disease, ischemic stroke, pulmonary hypertension, indicating that MSC-derived EVs exert similar effects as MSCs [64,65,66,67,68]. Due to their high stability in the bloodstream and the capacity to penetrate blood-brain barrier (BBB), MSC-derived EVs have a great potential for the treatment of neurological and neurodegenerative diseases, which has been experimentally confirmed as EVs administration through both intravenous and intranasal routes [25,69,70,71,72]. Additionally, an inflammatory state in AD or Parkinson’s disease (PD) makes the BBB more vulnerable to facilitate EVs transport from the peripheral circulation to the brain [73,74,75]. Therefore, it could reasonably be expected that MSC-derived EVs manifest beneficial effects in AD treatments.

MSC-derived EVs have shown promise in improving the cognitive deficits induced by Aβ_1–42_ aggregates and promoting neurogenesis in the hippocampus and subventricular zone (SVZ), which are of great significance in the transition from short-term memory to long-term memory [70,76]. In vitro results have addressed that MSC-derived EVs protect neurons from oxidative stress and synapse damage induced by Aβ oligomers [77,78]. Wang et al. and our lab demonstrated the positive effect of using MSC-derived EVs (BM-MSCs and WJ-MSCs, respectively) both in vitro and in vivo [71,79]. BM-MSC-derived EVs significantly reduced Aβs induced inducible nitric oxide synthase (iNOS) expression in cultured primary neurons. Administration of BM-MSC-derived EVs intracerebroventricularly was shown to improve cognitive behavior, rescue synaptic transmission in hippocampal CA1 regions and long-term potentiation (LTP) in APP/PS1 transgenic mice. In our research, the human neuroblastoma cell line overexpressing FAD mutations and J20 transgenic mice were used to investigate the therapeutic effect of WJ-MSC-derived EVs. Reduced Aβ expression and restored expression of neuronal memory/synaptic plasticity-related genes were observed in the cell model. In vivo studies demonstrated improved cognitive function, restored glucose metabolism, and inhibited astrocytes/microglia activation in mice that were administrated with WJ-MSC-derived EVs through an intravenous injection [71]. Furthermore, an alternative delivery of MSC-derived EVs for therapeutic intervention in AD through the intranasal route has been used in recent studies owing to the safety, low invasive procedure and a higher amount of EVs reaching the brain [70,80,81]. Similarly, MSC-derived EVs exhibit neuroprotective and immunomodulatory potential, evidenced by increased dendritic spine density and decreased microglia activation in treated mice [80]. Another recent study has demonstrated that MSC-derived EVs can lower Aβ plaque burden and decrease the colocalization between Aβ plaque and glial fibrillary acidic protein (GFAP, a reactive astrocyte marker) in the brain [81]. The therapeutic effects of MSC-derived EVs obtained from the cell and animal models of AD are summarized in Table 1.

## 5. Therapeutic Mechanisms of MSC-Derived EVs Actions in AD

Accumulating studies have uncovered that the considerable therapeutic benefits of MSC-derived EVs can be attributed to the ability to degrade Aβs, modulate immunity and protect neurons in the brain (Figure 1). In this context, MSC-derived EVs are considered to be ideal potential therapeutics for AD.

### 5.1. Aβ Degradation

The Aβ plaques are composed of Aβ peptides, a 40–42 amino acids proteolytic fragment of amyloid precursor protein (APP) [93]. These Aβ peptides undergo an aggregation process resulting in the formation of soluble oligomeric species and insoluble fibrillar species, eventually ending with the deposition of plaques [94]. Numerous reports have described that the excess accumulation might be of a result of a metabolic imbalance between the production and clearance of Aβs, thereby triggering synaptic deficits, neuronal alterations and neurodegeneration [95,96,97]. In clearance systems of the brain, Aβ related degradation clearance is contributed by different proteases, such as neprilysin (NEP), matrix metalloproteinases (MMPs), and glutamate carboxypeptidase II [98,99]. Among them, the critical role of NEP in AD has been intensively studied and thus regarded as a potential target for the treatment of AD [100,101]. Moreover, the expression and activity of NEP are significantly reduced in patients with AD [102]. Enzymatically active NEP expressed in AD-MSC-derived EVs was suggested to decrease both extracellular and intracellular Aβ levels in the N2a cells (a mouse neuroblastoma cell line) [82]. In our study, WJ-MSCs-derived EVs also expressed active NEP on their membranes by means of Western blot and NEP-specific activity assay [71]. Furthermore, MSCs-derived EVs-treated AD rodent models exhibited elevated NEP and IDE expressions along with decreased Aβ depositions [85,92]. Taken together, these reports have demonstrated the potential of MSCs-derived EVs in the treatment of AD and further reflect the feasibility to lower brain Aβ levels by delivering NEP or other Aβ-degrading enzymes.

### 5.2. Neuroprotection and Neuroregneration

Neuronal networks, astrocytes, microglia and oligodendrocytes contribute to a complex cellular phase of AD evolving over decades. In view of the critical role of neurons in CNS, dysfunction of the brain with AD is mediated by reduction in synaptic plasticity, changes in homeostatic scaling and disruption of neuronal connectivity, which characterize AD dementia [103]. The neuroprotection and neurogenesis contributed by MSC-derived EVs have been demonstrated in vitro and in vivo as addressed above; some of them have delineated the mechanisms of MSC-derived EVs actions. De Godoy et al. reported that the catalase contained in MSC-derived EVs was responsible for neuroprotection from AβOs-induced oxidative stress, and the capacity was checked by a membrane-permeant specific catalase inhibitor [77]. Our study addressed that one potential mechanism of the upregulation of neuronal memory/synaptic plasticity-related genes was in part due to the epigenetic regulation of a class IIa histone deacetylase [71]. On the other hand, EVs isolated from hypoxia preconditioned MSCs culture medium were found to increase the level of miR-21 in the brain of treated AD mice. The replenishment of miR-21 restored the cognitive deficits in AD mice, suggesting that miR-21a act as a regulator in this process [86]. Additionally, in a rat model of traumatic brain injury, MSC-derived EVs transferred miR-133b into astrocytes and neurons to enhance neurogenesis and improve functional recovery [104]. Thus, understanding the detailed mechanisms of MSC-derived EVs actions involved in neuroprotection and neuroregneration is beneficial to enhance the therapeutic potential in AD.

### 5.3. Immunomodulation

Increasing evidence suggests that AD pathogenesis is closely associated with the neuroinflammation, which might occur at early stage or mild cognitive impairment (MCI) even before Aβ plaque formation [105,106]. MSC-based therapy has been widely conducted in various disease treatments based on their ability to limit tissue inflammation microenvironments through the release of immunomodulatory factors such as prostaglandin E2 (PGE2), hepatic growth factor (HGF), transforming growth factor-β (TGF-β), indolamine 2,3-dioxygenase-1 (IDO-1), interleukin-10 (IL-10) and nitric oxide [65]. In terms of MSC-derived EVs, they acquire a lot of immunologically active molecules to regulate immune cells and thus exert similar therapeutic effects to their parental MSCs [107]. As evidenced by Harting and colleagues, MSCs exposed to TNF-α and IFN-γ generated EVs with a distinctly different profile, including the protein and nucleic acid composition. These EVs were found to partially alter the COX2/PGE2 pathway to enhance their anti-inflammatory properties [108]. In the recent research, cytokine-preconditioned MSC-derived EVs were intranasally administrated into AD mice and found to induce immunomodulatory and neuroprotective effects, evidenced by the inhibition of microglia activation and an increment in the dendritic spine density [80]. Given that EVs isolated from cytokine-pretreated MSCs exhibit more remarkable anti-inflammatory abilities than naïve EVs, it implies that preconditioned MSC-derived EVs might be a better option in the treatment of AD or other inflammatory diseases. It is necessary to compare the therapeutic effect on AD between cytokine-preconditioned MSC-derived EVs and naïve MSC-derived EVs.

MSC-derived EVs can regulate enzyme activity to suppress inflammatory response. Aβs-induce iNOS in glial cells and the subsequent release of high levels of nitric oxide (NO) inhibit integrated mitochondrial respiration, resulting in cell death [109]. Wang et al. demonstrated that BM-MSC-derived EVs not only reduced the expression of iNOS in cultured primary neurons but also significantly alleviated the deficits of CA1 synaptic transmission in APP/PS1 mice [79]. In a similar manner, BM-MSC-derived EVs were able to decrease iNOS expression in a model of osteoarthritis [110]. Additionally, levels of inflammatory cytokines, including IL-1β, IL-6 and TNF-α, were also decreased after MSC-derived EVs treatment [80,92].

Plenty of studies mainly focus on the status of microglia regulated by MSC-derived EVs. In line with other findings, our study also showed that MSC-derived EVs inhibited astrocytes and microglia activation in the brain of AD mice, indicating that these effects are attributed to immunomodulatory properties of EVs [70,71,81]. It should be noted that neuronal networks, astrocytes microglia, oligodendrocytes and the vascular system all contribute to a complex cellular phase of the disease. Once the cellular homeostasis is no longer maintained, the clinical phase of AD is initiated [111]. MSC-based therapy is considered to exert a dynamic homeostatic response that assists in tissue preservation, as well as function recovery, as do the MSC-derived EVs [108,112]. Thus, the effect of MSC-derived EVs on oligodendrocytes and vascular system involved in AD pathogenesis is worthy of further investigation.

## 6. Clinical Trials of MSC-Derived EVs in AD

The concept of using MSC-derived EVs as a regenerative medicine for neurological diseases or conditions is relatively new. Despite that the results obtained from cell and mouse models of AD have suggested that MSC-derived EVs therapies are promising, few clinical studies for AD currently registered in the National Institutes of Health clinical trials database (Table 2). To date, only one clinical trial has been approved to explore the safety and effectiveness of MSC-derived EVs in patients with mild to moderate dementia (NCT04388982). The researchers plan to give patients three doses of ADSC-derived EVs (5, 10 and 20 μg) via nasal drip, twice a week for 12 weeks. Besides the measurements of liver or kidney function and treatment-related adverse events for safety, the cognitive function tests, quality of life, MRI and PET neuroimaging, and Aβ levels in serum and CSF are further evaluated by schedule. As a good example, the clinical trial of MSC-derived EVs therapy for acute ischemic stroke is implemented based on the finding of EVs mediated delivery of miR-124 inducing neurogenesis after ischemia (NCT03384433) and this study will be completed in December 2021 [113]. Of note, emerging clinical trials using MSC-derived EVs in the treatment of COVID-19 or viral pneumonia are planned for the next two years (NCT04276987, NCT04491240, NCT04657458, NCT04493242), which emphasizes the role of immunomodulation and regeneration of MSC-derived EVs in various diseases.

Although using MSC-derived EVs as cell-free therapy is promising, several major issues should be addressed. The safety and doses for clinical use are certainly priorities; other issues include the establishment of the optimal cell culture conditions, the protocol for isolation, characterization and quantification of MSC-derived EVs, and therapeutic schedules [10,114].

## 7. Strategies for EV-Based Therapies

As nanoscale biomaterials, various molecules including proteins, RNAs, DNAs, hydrophilic and hydrophobic drugs have been successfully loaded into EVs [115]. The effect induced by these powerful agents in MSC-derived EVs are considered to reflect the “homing” ability of their parental cells, therefore, several studies have demonstrated that the lesions can attract MSC-derived EVs to their vicinity [116,117]. Generally, EVs tend to accumulate in organs that belong to mononuclear phagocyte system (MPS) such as liver and spleen, whereas the uptake by other organs is much lower and the clear-up is much faster [118,119,120]. In the brain, the accumulation of MSC-derived EVs is hardly found due to the BBB. Although MSC-derived EVs with the property of homing should pass through the tight junction in a pathological state expectedly, it is still necessary to enhance the efficacy by increasing the uptake and accumulation in the brain. Several approaches have been investigated to broaden or enhance their therapeutic properties through the modification of EVs and the route of administration applied (Table 3).

EVs can be chemically or biologically modified to insert membrane-binding species (e.g., peptides) into the membrane surface and package materials (e.g., drug, miRNA or small interfering RNA) into their vesicle interior [121,122,123]. To improve the brain targeting ability of MSC-derived EVs, Alvarez-Erviti et al. engineered dendritic cells to express Lamp2b, a membrane protein in EVs, fused to the neuron-specific rabies viral glycoprotein (RVG) peptide [69]. These RVG-tagged EVs were additionally loaded with exogenous siRNA specific to β-secretase 1 (BACE1), a key protease implicated in Aβ production. These modified EVs manifested therapeutic potential in AD therapy, demonstrated by the delivery of siRNA specifically to neurons, microglia, oligodendrocytes in the brain and the strong knockdown of BACE1. In a similar manner, this strategy has been applied for guiding MSC-derived EVs to the brain to alleviate AD pathology and deliver certain nucleic acids (DNA aptamer or shRNA specific to α-synuclein) to reduce the α-synuclein aggregates in the PD model [87,124,125]. The utilization of RGD peptides led the EVs to pass the BBB and target the ischemic lesion; meanwhile, the curcumin loaded in EVs was successfully delivered to repress the inflammatory response and cellular apoptosis [126]. Furthermore, EVs can be used as a theranostic agent, combining both targeted imaging and therapeutic effects. Jia et al. loaded superparamagnetic iron oxide nanoparticles (SPIONs) and curcumin into the EVs conjugated with neuropilin-1-targeted peptide and subsequently found the diagnostic and therapeutic effects on glioma were significantly enhanced [127].

To our knowledge, robust neurogenesis was observed after MSC-derived EVs treatment primarily via intracerebroventricular injection or intranasal route in comparison with that via systemic route, suggesting that the amount of MSC-derived EVs accumulated in brain reflects the therapeutic efficacy [70,76,80,81,128]. In addition to intranasal delivery, it is feasible to induce permeability by a temporary disruption of the BBB, such as pulsed focused ultrasound (pFUS). Bai et al. reported that pFUS increased the homing of blood serum-derived EVs to the brain by 4.45-fold and thus the glioma growth significantly was suppressed [129]. Noteworthy, the increased homing to the brain might be attributed to the use of different sources of EVs that certainly express different homing factors on the surface membrane. For example, macrophage-derived EVs were characterized to express the integrin lymphocyte function-associated antigen 1 (LFA-1) and thus was able to interact with intercellular adhesion molecule 1 (ICAM-1), upregulated in inflammation, to promote the uptake of EVs in the BBB cells [130].

Therefore, based on the achievements of abovementioned studies in brain diseases, MSC-derived EVs not only can be applied as active drug themselves but also can be used as a drug delivery vehicle after exogenously re-engineering and modification.

## 8. Conclusions and Prospects

Taken together, MSC-derived EVs have a lot of potential as therapeutics for AD. In addition to the therapeutic effects, similar to their parent cells, concomitantly they have a lower risk of teratoma formation and the capacity to cross BBB. Currently, the comprehensive works regarding nomenclature, classification and characterization of EVs and their subgroups should be urgently integrated to accelerate research on EVs. The safety, toxicity and doses also need to be further investigated to support the development from bench to bedside. For the EVs’ industry, the address of issues such as the robustness of manufacture, uniformity of production, and scale-up of processes are their priority. In addition, the strategies in accelerating EVs delivery and the action mechanisms should be further clarified. The underlying cellular and molecular mechanisms could stimulate studies about the understanding of pathogenesis and the employment of therapeutic strategies for AD. Especially, the importance of non-neuronal cells in the brain affected by AD is unneglectable. Despite that the utilization of MSC-derived EVs in the treatment of AD is promising, the clinical translation remains a huge challenge and further studies should be carried out to tackle the complicated pathology and promiscuous signaling pathway of AD.

## Figures and Tables

**Figure 1 membranes-11-00796-f001:**
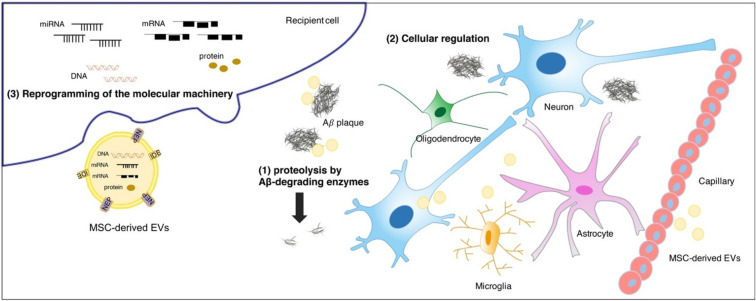
The illustration of potential mechanisms of MSC-derived EVs actions in AD. The therapeutic benefits of MSC-derived EVs are attributed to (**1**) the ability to degrade Aβs by membrane-bound Aβ-degrading enzymes, such as NEP and IDE; (**2**) the capability to regulate various cells in the brain including immunomodulation or neuroregeneration; (**3**) the reprogramming of the molecular machinery in recipient cells via proteins, mRNAs, and miRNAs transferred by EVs.

**Table 1 membranes-11-00796-t001:** A summary of preclinical studies of MSC-derived EVs-based therapy both in vitro and in vivo models of AD.

Model	Source of EVs	Protocol	Administration Route	Reported Effects	Ref.
**In vitro models**					
N2a cells	ADSCs	500 µg/well, 24 h	Co-culture	Decreased extracellular and intracellular Aβs levels	[82]
SH-SY5Y-APPswe cells	UC-MSC	2 μg/well for 24 h	Co-culture	Decreased extracellular and intracellular Aβs levels	[83]
SH-SY5Y-APP(S/L) cells	WJ-MSCs	50 µg/well, twice a week for 1 week	Co-culture	Decreased Aβs expression and restored the expression of neuronal memory/synaptic plasticity-related genes	[71]
NSCs isolated from Tg2576 mice	ADSCs	200 μg/mL for 24 or 48 h	Co-culture	Reduced Aβ levels and the Aβ 42/40 ratio, increased neurite growth and alleviated cell apoptosis	[84]
Cortical neuron culture from newborn APP/PS1 mice	BM-MSCs	100 μg/mL for 12 h	Co-culture	Reduced Aβs induced iNOS expression	[79]
Hippocampal neuron culture from rat embryos (E18)	BM-MSCs isolated from Wistar rats	2.4 × 10^8^ particles for 22 h	Co-culture (Pretreatment with 500 nM of AβOs for 2 h)	Protected neurons from AβOs-induced oxidative stress and synapse damage	[77]
Hippocampal neuron culture from rat embryos (E18)	WJ-MSCs	6 × 10^8^ particles for 22 h	Co-culture (With 500 nM of AβOs for 2 h)	Protected neurons from AβOs-induced oxidative stress and synapse damage	[78]
Cortical neurons culture from C57BL/6 mice embryos (E13–15)	ADSCs	0.05, 0.1, 1 μg/mL for 24 h	Co-culture (Pretreatment with 20 μM of AβOs)	Alleviated AβOs-induced neuronal toxicity	[70]
**In vivo models**					
APP/PS1 mice					
No age indicate	BM-MSCs	100 μg/5 μL, once per 2 days for 2 weeks	i.c.v.	Improved cognitive behavior, rescued impairment of CA1 synaptic transmission and LTP	[79]
7-month-old	UC-MSC	30 μg/100 μL, every 2 weeks, four times	i.v.	Reduced Aβ deposition, improved cognitive behavior; enhanced expression of IDE and NEP; modulated the activation of microglia	[85]
7-month-old	PC-BM-MSCs	150 μg/80 μL, biweekly for 4 months	i.v.	Improved cognitive behavior, reduced Aβ deposition; decreased proinflammatory factors and increased anti-inflammatory factors	[86]
7-month-old	RVG-BM-MSCs	5 × 10^11^ particles/100 μL, monthly for 4 months	i.v.	Improved cognitive behavior, reduced Aβ deposition, and restored the levels of inflammatory cytokines	[87]
5-month-old	BM-MSCs	22.4 μg/4 μL	i.c.v	Reduced Aβ deposition and the amount of dystrophic neurons in both the cortex and hippocampus	[88]
9-month-old	UC-MSC	2 mg/mL, continuously at 0.25 µL/h for 14 days	i.c.v.	Reduced Aβ deposition, improved cognitive behavior and inhibited the inflammatory and oxidative stress	[83]
7-month-old	BM-MSCs	50 μg/80 μL, every 2 weeks for 16 weeks	i.v.	Reduced Aβ deposition, promoted cognitive function recovery and increased NeuN expression	[89]
4-month-old	miRNA-22-loaded mouse ADSCs	100 μg/mL, every 7 days until 30 days	i.v.	Improved cognitive behavior, inhibited the inflammatory factors expression and reduced the nerve cell damage	[90]
9-month-old	ADSCs	1 mg/kg in 10 μL, every two days for 2 weeks	IN	Ameliorated neurologic damage in the whole brain areas, increased neurogenesis, reduced Aβ deposition and decreased microglia activation	[70]
J20 mice					
9-month-old	WJ-MSCs	50 µg/100 µL, once a week for 4 weeks	i.v.	Restored the expression of neuronal memory/synaptic plasticity-related genes, improved brain glucose metabolism and cognitive function; inhibited astrocyte and microglia activation	[71]
3 × Tg					
7-month-old	Cytokine-preconditioned BM-MSCs	30 μg/100 µL	IN	Decreased microglia activation and increased dendritic spine density	[80]
5 × FAD					
2-month-old	BM-MSCs	20 × 10^8^ particles in 5 µL every 4 days until 4 months of age	IN	Improved cognitive behavior, reduced Aβ deposition in the hippocampus and decreased colocalization between GFAP and Aβ plaques	[81]
1.5–2.5-month-old 5.0–6.5-month-old	hNSC	2.25 × 10^7^ particles in 50 μL hibernation buffer	i.v. via RO injection	Restored fear extinction memory consolidation and reduced anxiety related behaviors; reduced the dense core Aβ plaque number and microglial activation; restored synaptophysin in the AD brain and homeostatic levels of pro-inflammatory cytokines	[91]
Administration of Aβ peptides into the dentate gyrus of C57BL/6 mice 8-week-old	MSCs (No source indicated)	10 µg/2 µL of PBS	i.c.v. into the dentate gyrus	Promoted neurogenesis in the SVZ and alleviated Aβ_1–42_-induced cognitive impairment	[76]
Administration of Aβ peptides into the lateral ventricle of SD rats (7-week-old)	BM-MSCs isolated from SD rats	30 μg/100 µL, once a month for 2 months	i.c.v. into the lateral ventricle	Reduced Aβ deposition, reduced the levels of inflammatory cytokines, elevated NEP and IDE expressions, increased neuron viability and reduced apoptosis rate	[92]

Abbreviation: ADSC, adipose tissue-derived mesenchymal stem cells; WJ-MSCs, Wharton’s jelly mesenchymal stem cells; UC-MSC, umbilical cord mesenchymal stem cells; BM-MSCs, bone marrow-derived mesenchymal stem cells; PC, hypoxia-preconditioned; NSCs, neuronal stem cells; SD rat, Sprague–Dawley rat; RVG, rabies viral glycoprotein; i.v., intravenous injection; i.c.v, intracerebroventricular injection; IN, intranasal; RO, retro-orbital sinus; AβOs, Aβ oligomers; LTP, long-term potentiation; SVZ, subventricular zone; GFAP, glial fibrillary acidic protein; NEP, neprilysin; IDE, insulin-degrading enzyme.

**Table 2 membranes-11-00796-t002:** Clinical trials using MSC-derived EVs in therapies.

Disease/Condition	Clinical Trial Number	Title	Sponsor
“Exosome” used in title			
Alzheimer’s disease	NCT04388982	The Safety and the Efficacy Evaluation of Allogenic Adipose MSC-Exos in Patients with Alzheimer’s Disease (Adipose MSC-derived exosomes)	Ruijin Hospital
Cerebrovascular disorders	NCT03384433	Allogenic Mesenchymal Stem Cell Derived Exosome in Patients with Acute Ischemic Stroke	Isfahan University of Medical Sciences
Acute respiratory distress syndrome (ARDS)	NCT04602104	A Clinical Study of Mesenchymal Stem Cell Exosomes Nebulizer for the Treatment of ARDS	Ruijin Hospital
Coronavirus	NCT04276987	A Pilot Clinical Study on Inhalation of Mesenchymal Stem Cells Exosomes Treating Severe Novel Coronavirus Pneumonia	Ruijin Hospital
Healthy	NCT04313647	A Tolerance Clinical Study on Aerosol Inhalation of Mesenchymal Stem Cells Exosomes in Healthy Volunteers	Ruijin Hospital
Macular holes	NCT03437759	MSC-Exos Promote Healing of MHs	Tianjin Medical University
Multiple organ failure	NCT04356300	Exosome of Mesenchymal Stem Cells for Multiple Organ Dysfuntion Syndrome After Surgical Repaire of Acute Type A Aortic Dissection	Fujian Medical University
Dry eye	NCT04213248	Effect of UMSCs Derived Exosomes on Dry Eye in Patients With cGVHD (Umbilical MSCs derived exosomes)	Zhongshan Ophthalmic Center, Sun Yat-sen University
Drug-resistant	NCT04544215	A Clinical Study of Mesenchymal Progenitor Cell Exosomes Nebulizer for the Treatment of Pulmonary Infection	Ruijin Hospital
Sepsis Critical illness	NCT04850469	Study of MSC-Exo on the Therapy for Intensively Ill Children	Children’s Hospital of Fudan University
Periodontitis	NCT04270006	Effect of Adipose Derived Stem Cells Exosomes as an Adjunctive Therapy to Scaling and Root Planning in the Treatment of Periodontitis: A Human Clinical Trial (Adipose derived stem cells exosomes)	Beni-Suef University
COVID-19 SARS-CoV-2 Pneumonia	NCT04491240	Evaluation of Safety and Efficiency of Method of Exosome Inhalation in SARS-CoV-2 Associated Pneumonia	State-Financed Health Facility “Samara Regional Medical Center Dinasty”
“Extracellular vesicle” used in title			
Bronchopulmonary Dysplasia	NCT03857841	A Safety Study of IV Stem Cell derived Extracellular Vesicles (UNEX-42) in Preterm Neonates at High Risk for BPD	United Therapeutics
Dystrophic epidermolysis Bullosa	NCT04173650	MSC Evs in Dystrophic Epidermolysis Bullosa	Aegle Therapeutics
COVID-19 ARDS Hypoxia Cytokine storm	NCT04657458	Expanded Access Protocol on Bone Marrow Mesenchymal Stem Cell Derived Extracellular Vesicle Infusion Treatment for Patients With COVID-19 Associated ARDS (BM-MSC derived EVs)	Direct Biologics, LLC
COVID-19 ARDS Pneumonia, Viral	NCT04493242	Extracellular Vesicle Infusion Treatment for COVID-19 Associated ARDS (BM-MSC derived EVs)	Direct Biologics, LLC

According to clinicaltrials.gov as of 30 August 2021.

**Table 3 membranes-11-00796-t003:** A summary of strategies that enhance the efficacy of EV-based therapy for brain diseases.

Strategies	Cargo-Loaded Molecules	Source of EVs	Disease	Reported Effects	Ref.
**Peptide-tagged**					
Rabies viral glycoprotein (RVG)	Nucleic acid siRNAs specific to BACE1	Dendritic cells	AD	Significant knockdown of BACE1 in mRNA and protein levels	[69]
	Naturally production	BM-MSCs	AD	Improved cognitive behavior, reduced Aβ deposition, and restored the levels of inflammatory cytokines	[87]
	miR-124	Mouse BM-MSCs	Ischemic stroke	Promoted cortical neurogenesis	[113]
	siRNAs specific to α-synuclein	Dendritic cells	PD	Decreased α-synuclein aggregation and rescued the loss of dopaminergic neurons	[124]
	DNA aptamers that recognize the α-synuclein	HEK293T	PD	Reduced α-synuclein aggregation and improved motor impairments	[125]
RGD peptides	Drug loaded Curcumin	Mouse BM-MSCs	Ischemic stroke	Strong suppression of the inflammatory response and cellular apoptosis	[126]
T7 peptide	Antisense miRNA oligonucleotides against miR-21 (AMO-21)	HEK293T	Glioblastoma	Reduction of tumor sizes	[131]
NRP-1-targeted RGE peptide	Superparamagnetic iron oxide nanoparticles (SPIONs) and curcumin	Raw264.7 cells, a macrophage cell line	Glioma	Delayed tumor recurrence, extended the survival of tumor-bearing mice and had targeted-imaging ability	[127]
Low-density lipoprotein (LDL)	Drug loaded Methotrexate	L929, a mouse fibroblastic cell line	Glioma	Prolonged the median survival period	[132]
**Natural production**					
LFA-1 expression	BDNF	Macrophage	PD	Enhanced delivery and accumulation in inflamed brain	[130]
Unidentified	Paclitaxel and doxorubicin	Brain endothelial cells	Brain cancer	Induction of cytotoxic effects against brain cancer	[133]
**Administration route**					
IN	Unmodified	ADSCs	AD	Decreased AβOs-induced neuronal toxicity	[70,81]
	Cytokine-stimulated	BM-MSCs	AD	Increased dendritic spine density, reduced Aβ deposition and microglia activation	[80]
	Drug loaded Curcumin, JSI-124, a Stat3 inhibitor	EL-4, a T cell line	Inflammation-mediated disease models, including LPS-induced brain inflammation model, EAE model and a GL26 brain tumor model	Selectively taken up by microglia and induced apoptosis	[134]
Disruption of BBB by pFUS	Unmodified	Blood serum	Glioma	Suppressed glioma growth with no obvious side effects	[129]

Abbreviation: IN, intranasal; NRP-1, Neuropilin-1; LFA-1, lymphocyte function-associated antigen 1; ICAM-1, intercellular adhesion molecule 1; pFUS, pulsed focused ultrasound; BACE 1, Beta-secretase 1; Stat3, signal transducer and activator of transcription 3; BDNF, brain-derived neurotrophic factor; LPS, lipopolysaccharide; EAE, experimental autoimmune encephalitis.

## Data Availability

Not applicable.

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
