# Peer review of "Mesenchymal Stem Cell-Derived Extracellular Vesicle-Based Therapy for Alzheimer’s Disease: Progress and Opportunity"

_membranes, 2021, doi:10.3390/membranes11100796_

Round 1

Reviewer 1 Report

This review aims to present recent progress on the therapeutic potential of EVs derived from mesenchymal stem cells (MSCs) in the treatment of AD, yet only 38% of the cited references are from 2019-2021 period. I believe at least 50% would be more appropriate to offer a perspective on the newest discoveries (e.g., Adv. Drug Deliv. Rev. 174, 535-552, 2021; Sci. Rep. 11, 18518, 2021; Alz. Res. Therapy 13, 57, 2021)

Reviewer 2 Report

In the manuscript, the authors discuss the perspectives of using mesenchymal stem cell-derived extracellular vesicles for treatment or alleviation of Alzheimer’s Disease. The introduction gives a detail-rich overview of the molecular remedies currently used or under development.

The EVs characteristics and recommendations for characterization are further on described.   The currently known action mechanisms are discussed and the manuscript includes a table with current preclinical studies toward using stem cell-derived EVs in the treatment of AD. Aß plaques increased clearance mechanism and/or supposed implication of EVs, the neuroprotection and immunomodulation are three sets of data presented that build a panel of molecular mechanisms in which the stem cells, including mesenchimal stromal cells, may be involved. A chapter describing current clinical studies and strategies for EV therapies ends the review.

The manuscript is well balanced in the structure of chapters and in the weight of specific data, both medical and cellular mechanisms. An in-depth systematization of the best current practices, based on the results of the preclinical and clinical studies and/or on type of stem cells/EVs characterization would be a benefit of chapter 7.

Anyhow, the first clinical trial presented in table 2 is the only example EVs (precisely exosomes) tested for treatment of AD. Hence, the conclusion of the review seems to be bold (lines 401-402, 412-414).

Lines 81-82 would need a citation.
